# Denoising Vanilla Autoencoder for RGB and GS Images with Gaussian Noise

**DOI:** 10.3390/e25101467

**Published:** 2023-10-20

**Authors:** Armando Adrián Miranda-González, Alberto Jorge Rosales-Silva, Dante Mújica-Vargas, Ponciano Jorge Escamilla-Ambrosio, Francisco Javier Gallegos-Funes, Jean Marie Vianney-Kinani, Erick Velázquez-Lozada, Luis Manuel Pérez-Hernández, Lucero Verónica Lozano-Vázquez

**Affiliations:** 1Escuela Superior de Ingeniería Mecánica y Eléctrica Unidad Zacatenco Sección de Estudios de Posgrado e Investigación, Instituto Politécnico Nacional, Mexico City 07738, Mexico; amirandag1100@alumno.ipn.mx (A.A.M.-G.); fgallegosf@ipn.mx (F.J.G.-F.); evelazquezl@ipn.mx (E.V.-L.); lperezh1602@alumno.ipn.mx (L.M.P.-H.); llozanov0800@alumno.ipn.mx (L.V.L.-V.); 2Departamento de Ciencias Computacionales, Tecnológico Nacional de México, Cuernavaca 62490, Mexico; dante.mv@cenidet.tecnm.mx (D.M.-V.); jkinani@ipn.mx (J.M.V.-K.); 3Centro de Investigación en Computación, Instituto Politécnico Nacional, Mexico City 07738, Mexico; pescamillaa@ipn.mx; 4Unidad Profesional Interdisciplinaria de Ingeniería Campus Hidalgo, Instituto Politécnico Nacional, Pachuca de Soto 42162, Mexico

**Keywords:** denoising vanilla autoencoder, images, noise

## Abstract

Noise suppression algorithms have been used in various tasks such as computer vision, industrial inspection, and video surveillance, among others. The robust image processing systems need to be fed with images closer to a real scene; however, sometimes, due to external factors, the data that represent the image captured are altered, which is translated into a loss of information. In this way, there are required procedures to recover data information closest to the real scene. This research project proposes a Denoising Vanilla Autoencoding (*DVA*) architecture by means of unsupervised neural networks for Gaussian denoising in color and grayscale images. The methodology improves other state-of-the-art architectures by means of objective numerical results. Additionally, a validation set and a high-resolution noisy image set are used, which reveal that our proposal outperforms other types of neural networks responsible for suppressing noise in images.

## 1. Introduction

Currently, there is a growing interest in the use of artificial vision systems for application in daily tasks such as industrial processes, autonomous driving, telecommunication systems, surveillance systems, and medicine, among others [1]. Recent developments in the field of artificial vision have stimulated the need to make increasingly robust systems to meet established quality requirements, which is an essential part of why systems fail to cover these types of requirements, mainly in data acquisition. Among image acquisition systems, there are several factors that can alter the result of the capture, including failures in the camera sensors, adverse lighting conditions, electromagnetic interferences, noise generated by the hardware, etc. [2]. All of these phenomena are described using distribution models and are known, in a general way, as noise. The procedure in the image processing field to try to diminish the effect of the noise is known as the pre-processing stage in any image processing system. In recent years, various algorithms have been developed in denoising images, and recently, a new field has taken much interest in the scientific community. In this way, deep learning methods emerge [3,4].

Deep learning methods particularly present an inherent ability to overcome the deficiencies contained in some traditional algorithms [5]; however, despite their significant improvements compared to traditional filters, deep learning methods have practical limitations to their credit, which fall in high computational complexity. Although, as previously mentioned, various methods have focused on noise suppression, in this work, autoencoders are proposed, which are neural networks capable of replicating an unknown image by applying convolutions whose weights were adjusted with previous training [6,7,8]. This research project highlights the importance of using autoencoders because they do not require high computational complexity, demonstrating noticeable improvement compared to other types of deep learning architectures, such as the Denoising Convolutional Neural Network (*DnCNN*) [9], the Nonlinear Activation Free Network for Image Restoration (*NAFNET*) [10], and the Efficient Transformer for High-Resolution Image Restoration (*Restormer*) [11].

The rest of this paper is structured as follows. In Section 2, the theoretical background work is described. The proposed model is described in Section 3. The experimental setup and results are discussed in Section 4. Finally, the conclusions of this research work are given in Section 5.

## 2. Background Work

In recent years, noise suppression has become a dynamic field within the domain of image processing. This is due to the fact that as technological advances emerge, a greater understanding of the scene in which a vision system is interacting is required [12]. For the suppression of noise, several processing techniques have been proposed. These techniques are known as filters that depend on the noise present in the image and are mainly classified into two types.

### 2.1. Spatial Domain Filtering

Spatial filtering is a traditional method for noise suppression in images. These filters suppress noise by being applied directly to the corrupted image. They can generally be classified into linear and non-linear. Among the most common filters are:Mean Filter: For each pixel, there are samples with a similar neighborhood to the pixel’s neighborhood, and the pixel value is updated according to the weighted average of the samples [13].Median Filter: The use of this filter is that the central pixel of a neighborhood is replaced by the median value of the corresponding window [14].Fuzzy Methods: This type of filter is different from those mentioned above since it is mainly constituted by fuzzy rules with which it is possible to preserve the edges and fine details in an image. Fuzzy rules are used to derive suitable weights for neighboring samples by considering local gradients and angle deviations. Finally, directional processing is used with which it improves the precision of the same filter [15].

### 2.2. Transform Domain Filtering

Transform domain filtering is a very useful tool for signal and image processing due to its extensive analysis of multiple resolutions, sub-bands, and location in the time and frequency domains. An example of this type of filtering is the Wavelet method, which is performed based on the frequency domain and attempts to distinguish the signal from noise and preserve said signal in the noise suppression process. As a first step, a wave base is selected to determine the decomposition of its layers to later select the level of decomposition, establishing a threshold in all the sub-bands for all levels [16].

### 2.3. Artificial Intelligence

A new method of processing images has emerged, called artificial intelligence. To address the issue of noise suppression, it is necessary to distinguish between artificial intelligence, machine learning, and deep learning, because people tend to use these terms synonymously, but there exists a subtle difference. Artificial intelligence involves machines that can perform tasks with characteristics of human intelligence, such as understanding language, recognizing objects, gestures, sounds, and problem solving [17,18]. Machine learning is a subset that belongs to artificial intelligence. The function is to obtain better performance in the learning task. The algorithms used are mainly statistical and probabilistic ones, making the machines improve with experience, allowing them to act and make decisions based on the input data [19]. Finally, deep learning is a subset of machine learning that uses techniques and algorithms of automatic learning that have high performance in different problems of image recognition, sound recognition, etc., since the basic functioning and structure of the brain and the visual system of animals are imitated [20].

There are two types of deep learning: the first type is supervised, learning which takes a direct approach using labels on learning data to build a reasonable understanding of how machines make decisions, and the second is unsupervised learning, which takes a very different approach by learning by itself how to make decisions or perform specific tasks without the need to contain labels in a database [21].

#### Autoencoders

Autoencoders are unsupervised neural networks, and the main function of autoencoders is that the input and the output are the same [22]. This is taken as an advantage against other models because, in each training phase of the neural network, the output is compared with the original image version, and through a calculation error, the weights found in each of the layers that make up the autoencoder are adjusted. This adjustment is carried out by means of the backpropagation method. There are different types of autoencoders, which are:The Vanilla Autoencoder (*VA*) comprises only three layers: the encoding layer, in charge of reducing the dimensions of the input information; the hidden layer, better known as latent space, in which are the representations of all characteristics learned by the network; and the decoding layer, which is in charge of restoring the information to its original input dimensions, as shown in Figure 1 [23].The Convolutional Autoencoder (*Conv AE*) makes use of convolution operators and extracts useful representations from the input data, as shown in Figure 2. The input image is sampled to obtain a latent representation and is forced to learn that representation [24].The Denoising Autoencoder (*DA*) is a robust modification of *Conv AE* that changes the input data preparation. The information the autoencoder is trained in is divided into two groups: original and corrupted. In order for the autoencoder to learn to denoise an image, the corrupted information is sent to the input of the network to be processed. Once the information is in the output, it is compared with the original [25]. This type of autoencoder is capable of generating clean images from noisy images, ignoring the type of noise present as well as the density in which the image was affected.

## 3. Proposed Model

The proposed model is based on the suppression of Gaussian noise in both *RGB* and grayscale (*GS*) images. Figure 3 shows the architecture of the proposed Denoising Vanilla Autoencoder (*DVA*) algorithm, which consists of a selection stage where, if the image to which the processing is going to be submitted is of the *RGB* type, a multimodal model is applied, and if it is a *GS* image, a unimodal model is applied. This is described by Equation (Equation 1).

The advantage of combining two types of autoencoder architectures (*VA* and *DA*) is that by only having one encoding layer and one decoding layer, the reconstructed pixels do not have many alterations, which could translate into a loss of information, and at the same time, they are capable of remove noise present in images. The use of the autoencoder also allows us to have a lower computational load, which, in turn, improves both training and processing times once the network models are generated.
(1)$$X^{\prime }=\left\{\begin{array}{cccc} unimodal & c=1 & if & X_{\mathrm{GS}}\\ multimodal & c=3 & if & X_{\mathrm{RGB}} \end{array}\right.,$$
where $X^{\prime }$ is the image processed by DVA, and *c* is the number of channels in the corrupted image.
(2)$$X\in R^{w,h,c},W\in R^{m,n,c,k},$$
where X is the corrupted image with dimensions width *w*, height *h*, and channels *c*, and W is the matrix weight with dimensions width *m*, height *n*, channels *c*, and *k* kernels.
(3)$${\left(X*W\right)}_{\left(i,j,c\right)}=\sum_{m}\sum_{n}\sum_{k}\left(x_{\left(i+m-2,j+n-2,c\right)}\cdot w_{\left(m,n,c,k\right)}\right)+b_{c},$$
where ${\left(X*W\right)}_{\left(i,j,c\right)}$ is the intensity of the result of the *k* convolutions in the position (i,j,c), *b* is the bias.
(4)$$Y_{\left(i,j,c\right)}=f{\left(X*W\right)}_{\left(i,j,c\right)}$$
where $Y_{\left(i,j,c\right)}$ is the result of the activation function ReLu *f* in the position (i,j,c).
(5)$$f=\left\{\begin{array}{ccc} 0 & for & Y_{\left(i,j,c\right)}<0\\ Y_{\left(i,j,c\right)} & for & Y_{\left(i,j,c\right)}\geq 0 \end{array}\right.,$$
(6)$$Z_{\left(i,j,c\right)}=max\left\{Y_{\left(i+p,j+q,c\right)},Y_{\left(i+1+p,j+q,c\right)},Y_{\left(i+p,j+1+q,c\right)},Y_{\left(i+1+p,j+1+q,c\right)}\right\}$$
where *Z* is the encoded image by maxpooling, $p=\left\{0,1,2,\cdots ,\frac{w}{2}-1\right\}$, and $q=\{0,1,2,\cdots ,\frac{h}{2}-1\}$ are the strides.
(7)$$Z_{\left(i,j,c\right)}^{\prime }=f{\left(Z*W{}^{\prime }\right)}_{\left(i,j,c\right)}$$
where $Z_{\left(i,j,c\right)}^{\prime }$ is the result of the second convolutional layer and activation function, and $W^{\prime }$ is another matrix weight.
(8)$$\left\{Y_{\left(i+p,j+q,c\right)}^{\prime },Y_{\left(i+1+p,j+q,c\right)}^{\prime },Y_{\left(i+p,j+1+q,c\right)}^{\prime },Y_{\left(i+1+p,j+1+q,c\right)}^{\prime }\right\}=Z_{\left(i,j,c\right)}^{\prime }$$
where $Y^{\prime }$ is the dencoded image by upsampling.
(9)$$X_{\left(i,j,c\right)}^{\prime }={\left(Y^{\prime }*W^{\prime \prime }\right)}_{\left(i,j,c\right)}$$
where $X^{\prime }$ is the final result of the processing, and $W^{\prime \prime }$ represents another matrix weight.

For the multimodal model, the image is separated into its three different components (red, green, blue), and each component is processed independently, with models trained for each type of channel (Equations (Equation 2)–(Equation 9)) so that once the result is obtained, the three new ones are concatenated. The components generate a new image in which the noise is smoothed out. Within the unimodal model, a single trained model is applied. The main reason why a multimodal model was trained for *RGB*-type images is because the noise, being completely random and defined by a Gaussian probability, means that each channel is affected differently. In this case, processing the three channels of the image in the same way can cause the final smoothing to not be carried out properly and contain a greater number of corrupted pixels. Figure 4a shows the original histogram of the Lenna image, and Figure 4b shows how the image behaves when corrupted with Gaussian noise with density σ=0.20. This example is perceived as the red channel tends to increase the intensity of its pixels, and in the case of both the green channel and the blue channel, their intensities tend to decrease.

The *DVA* process is described in detail in Algorithm 1. Once the processing through the *DVA* is finished, we analyze the histogram of the resulting image, which is shown in Figure 5, perceiving how the *DVA* restores the intensities of the pixels contained in each of the channels to a certain extent. In this sense, the *DVA* is capable of restoring the image; however, it is not an optimal processing due to the nature of the noise since the same noise causes significant loss of information in the images, which the *DVA* tries to bring closer to the images. The intensities of the corrupted pixels are an ideal panorama.
**Algorithm 1:** Process image using DVA.
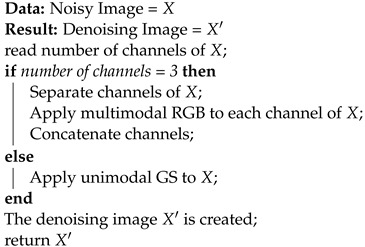


### Network Training

For the multimodal model, the “1 million faces” database was used, of which only 7000 images were used [26], which were resized in a dimension of 420 × 420 pixels. The same database was duplicated to generate the noise database. The 7000 images were divided into batches of 700 in which each batch was corrupted with a different noise density. The noise densities used are $\left\{0,0.1,0.15,0.2,0.25,0.3,0.35,0.4,0.45,0.5\right\}$. Once the two databases were obtained, the *DVA* training was carried out. The databases were divided into 80% for the training phase and 20% for the validation phase. In the case of the unimodal model, the original database was converted to *GS*, and the database with noise was created by repeating the above procedure.

The network was trained on an NVIDIA GeForce RTX 3070 (8GB) GPU. The hyperparameters used were seed = 17, learning rate = 0.001, shuffle = true, optimizer = Adam, loss function = MSE, epochs = 100, batch size = 50, and validation split = 0.1. Figure 6 shows the learning curves for the training and validation phase throughout the 100 epochs, showing us that the proposed architecture did not suffer from overtraining for both the unimodal model (Figure 6a) and the multimodal model (Figure 6b).

## 4. Experimental Results

The evaluation of the *DA* was carried out through the use of various images both in *RGB* and in *GS* of different dimensions. These images are unknown to the network in order to verify the proper functioning of the same. The evaluation images are shown in Figure 7. Each evaluation image was corrupted with Gaussian noise with densities from 0 to 0.50 in intervals of 0.01.

To gain a better perspective of the proper functioning of the proposed algorithm, comparisons were made with three other neural networks that differ in their structure but whose objective is noise smoothing. Table 1 shows the visual comparisons of the results obtained by the *DVA* and the other neural networks used to validate the algorithm for the Lenna image in *GS*. Table 2 shows the same comparisons for the Lenna image but this time in *RGB*. It should be noted that an approach was made to a region of interest to have a better perspective of the work of each of the networks on the image in question. In addition to the visual comparisons, evaluation metrics were used, such as:Mean Square Error (*MSE*): Calculate the mean of the differences between the original images and the processed images squared.
(10)$$MSE=\frac{1}{MN}\sum_{i=1}^{M}\sum_{j=1}^{N}{\left(x_{\left(i,j\right)}-y_{\left(i,j\right)}\right)}^{2},$$
where *x* and *y* are the images to compare, (i,j) is the coordinates of the pixel, and *M* and *N* are the size of the images.Root Mean Squared Error (*RMSE*): Commonly used to compare the difference between the original images and the processed images by directly computing the variation in pixel values [27].
(11)$$RMSE=\sqrt{\frac{1}{MN}\sum_{i=1}^{M}\sum_{j=1}^{N}{\left(x_{\left(i,j\right)}-y_{\left(i,j\right)}\right)}^{2}},$$Erreur Relative Globale Adimensionnelle de Synthèse (*ERGAS*): Used to compute the quality of the processed images in terms of normalized average error of each band of processed image [28].
(12)$$ERGAS=100\frac{dh}{dl}\sqrt{\frac{1}{n}\sum_{i=1}^{n}\frac{RMSE^{2}}{\mu_{i}^{2}}},$$
where $\frac{dh}{dl}$ is the ratio of pixel between hue and light, *n* is the number of bands, and $\mu_{i}$ is the mean of the *i*th band.Peak Signal-to-Noise Ratio (*PSNR*): A widely used metric that is computed by the number of gray levels in the image divided by the corresponding pixels in the original images and the processed images [29].
(13)$$PSNR=10log_{10}\frac{{\left(2^{b}-1\right)}^{2}}{\sqrt{MSE}},$$
where *b* is the number of the bits in the image.Relative Average Spectral Error (*RASE*): Characterizes the average performance of a method in the considered spectral bands [30].
(14)$$RASE=\frac{100}{\mu }\sqrt{\frac{1}{n}\sum_{i=1}^{n}\left(RMSE^{2}\right)\left(B_{i}\right)},$$
where μ is the mean radiance of the *n* spectral bands, and $B_{i}$ represents *i*th band of the image.Spectral Angle Mapper (*SAM*): Computes the spectral angle between the pixel, the vector of the original images, and the processed images [31].
(15)$$SAM=cos^{-1}\frac{\sum_{i=1}^{n}x_{\left(i,j\right)}y_{\left(i,j\right)}}{\sqrt{\sum_{i=1}^{n}x_{\left(i,j\right)}^{2}}\sqrt{\sum_{i=1}^{n}y_{\left(i,j\right)}^{2}}},$$Structural Similarity Index (*SSIM*): Used to compare the local patterns of pixel intensities between the original images and the processed images [32].
(16)$$SSIM=\frac{\left(2\mu_{x}\mu_{y}+C_{1}\right)\left(2\sigma_{xy}+C_{2}\right)}{\left(\mu_{x}^{2}+\mu_{y}^{2}+C_{1}\right)\left(\sigma_{x}^{2}+\sigma_{y}^{2}+C_{2}\right)},$$
where $\mu_{x}$ and $\mu_{y}$ are the mean of the images, respectively; $\sigma_{xy}$ is the covariance between the images to compare; $C_{1}={\left(k_{1}L\right)}^{2}$ and $C_{2}={\left(k_{2}L\right)}^{2}$ are two variables to stabilize the division with low denominators; *L* is the dynamic range of the pixel values; $K_{1}<<1$; and $K_{2}<<1$.Universal Quality Image Index (*UQI*): Used to calculate the amount of transformation of relevant data from the original images into the processed images [33].
(17)$$UQI=\frac{4\sigma_{xy}\mu_{x}\mu_{y}}{\left(\sigma_{x}^{2}+\sigma_{y}^{2}\right)\left(\mu_{x}^{2}+\mu_{y}^{2}\right)},$$

Table 3 exemplifies the *PSNR* results obtained by each neural network used in the validation *GS* images, and Table 4 exemplifies the *PSNR* results obtained in the same way but for *RGB* images.

In order to better show all the results of the metrics calculated from the validation database images processed by each of the aforementioned networks, Box-and-Whisker plots were made. This type of graph shows a summary of a large amount of data in five descriptive measures, in addition to intuiting its morphology and symmetry. This type of graph allows us to identify outliers and compare distributions.

Figure 8 shows the Box-and-Whisker plots for each of the metrics applied to the results of the *GS* images, and Figure 9 also shows the plots for the *RGB* image results. In each of the diagrams, it can be seen that the *DVA* contains smaller box dimensions with respect to the other networks, which means that the results obtained oscillate in a smaller range, so the result of the processing is similar regardless of the density with which the image is corrupted. The median is also located near the center of the box, which indicates that the distribution is almost symmetrical. Another point to highlight in the diagrams is that there are fewer outliers in the *DVA* compared to the other networks.

Recapitulating the previous results, it has been determined that the DVA obtained better results in comparison with the other neural networks. Although the difference presented in the metric calculations is not visually appreciated, this is mainly due to the fact that these metrics do not accurately reflect the perceptual quality of the human eye. One measure of image quality is the Mean Opinion Score (*MOS*) [34]; however, this type of measure is not objective as it differs depending on the user in question [35].

Another point in favor of the *DVA* is that it can be used in images of any dimension. As an example, Table 5 shows the visual and calculated results for high-definition images in which it is perceived that good restoration results are obtained.

As an aggregate, the negative of the differences between the analyzed image and the original image is shown, in which all the white pixels represent the pixels that are equal to those of the original image, for which it can be deduced that the *DVA* manages to have a good restoration of the image when it is corrupted with Gaussian noise.

## 5. Conclusions

In this research work, the importance of the use of filters for artificial vision systems was highlighted, as well as the basic concepts that encompass artificial intelligence and some types of unsupervised networks that are used today. Through this, a methodology based on autoencoders was proposed, which is capable of processing images of any size and type (RGB or GS). When carrying out the analysis of the results shown, it is identified that, from the use of the DVA, it is possible to efficiently smooth the Gaussian noise of images through the deep learning techniques implemented in the proposed algorithm regardless of the density of noise present in the corrupted images. The DVA results, both visual and calculated using various quantitative metrics, show better results in noise suppression compared to the DnCNN, NAFNET, and Restormer algorithms that, despite being of different architecture, have the function of smoothing noise in images.

One of the limitations observed during this research work is that when the image presents a low noise density, the results are similar to the architectures with which the DVA was compared. That is why it is suggested as a starting point to make improvements either by transferring learning or combining this methodology with another such as that proposed in [36] in order to obtain both qualitative and quantitative results, since it is extremely important for vision systems to get as close as possible to the real scene in order to reduce errors.

## Figures and Tables

**Figure 1 entropy-25-01467-f001:**

Architecture of the vanilla autoencoder.

**Figure 2 entropy-25-01467-f002:**
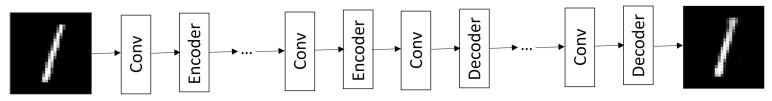
Architecture of the convolutional autoencoder.

**Figure 3 entropy-25-01467-f003:**
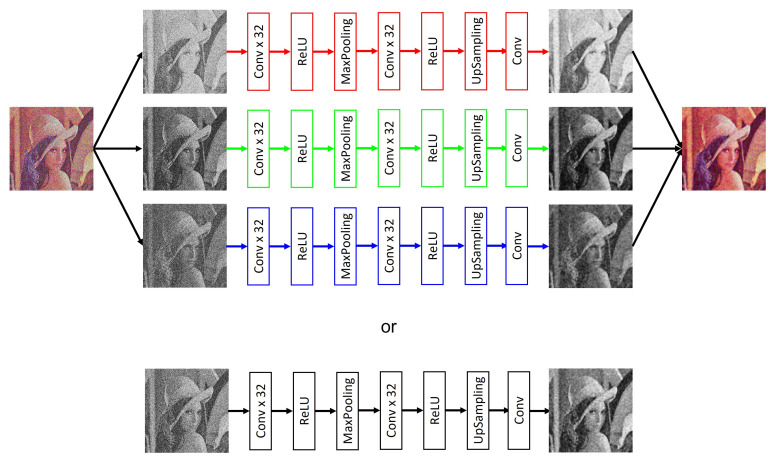
Architecture of the proposed denoising vanilla autoencoder.

**Figure 4 entropy-25-01467-f004:**
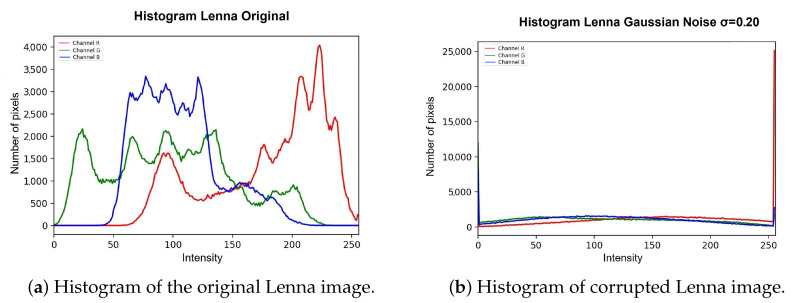
Difference between histogram of original Lenna image and histogram of corrupted Lenna image.

**Figure 5 entropy-25-01467-f005:**
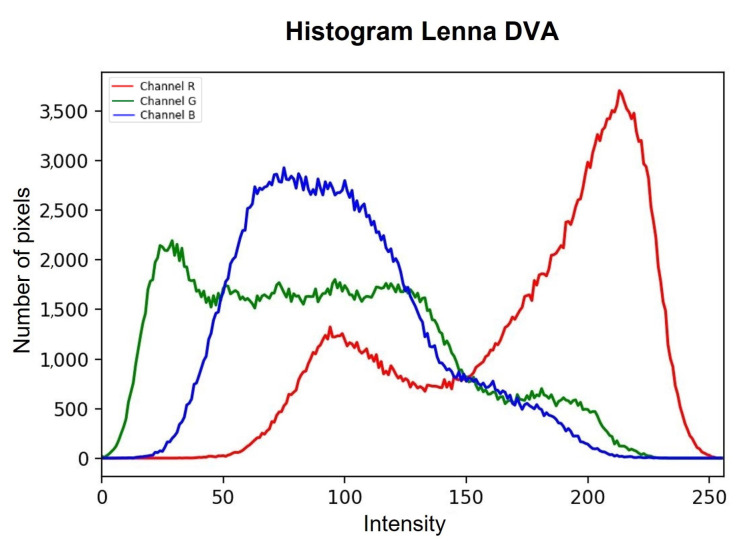
Histogram of the result of the corrupted image of Lenna processed by DVA.

**Figure 6 entropy-25-01467-f006:**
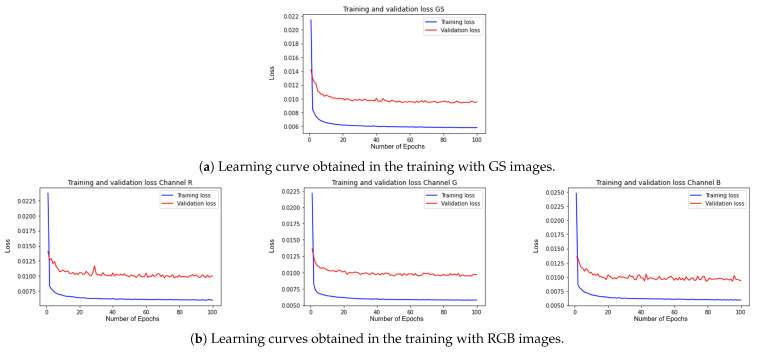
Learning curves obtained during the training of the DVA.

**Figure 7 entropy-25-01467-f007:**
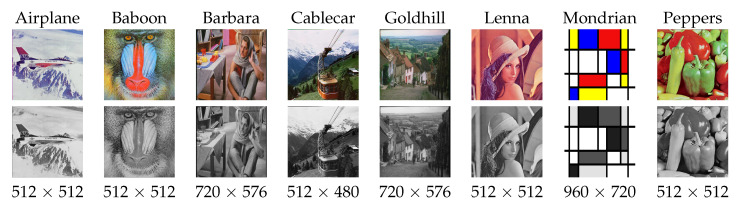
Testing images.

**Figure 8 entropy-25-01467-f008:**
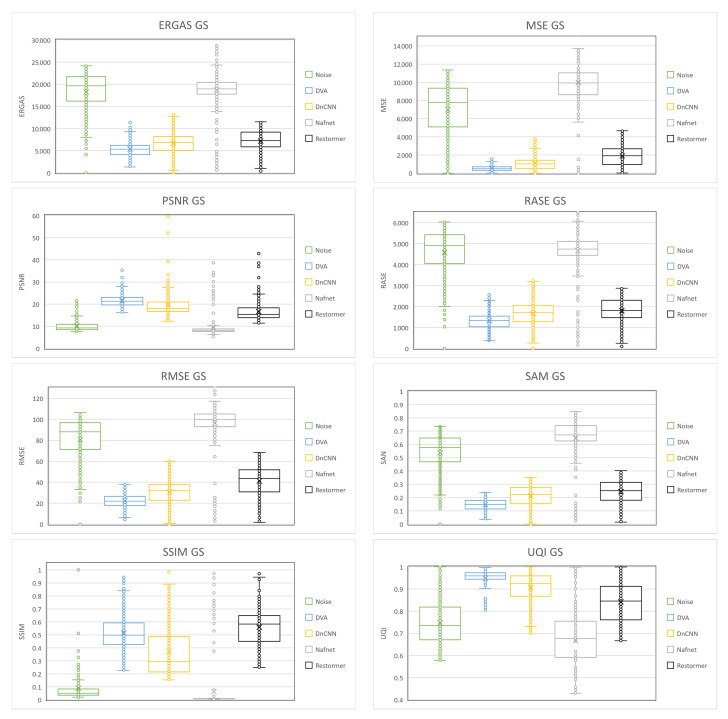
Box-and-Whisker plots of the quantitative results obtained on GS images.

**Figure 9 entropy-25-01467-f009:**
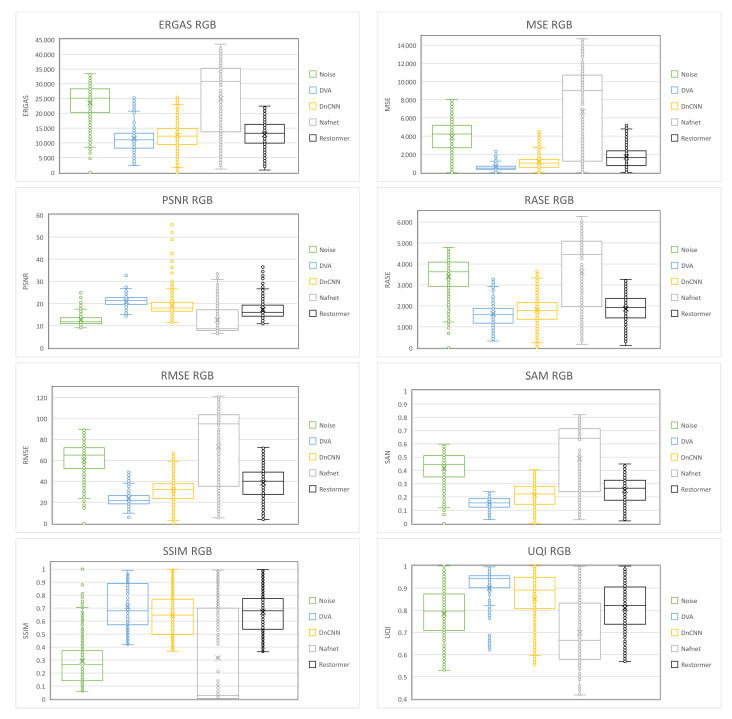
Box-and-Whisker plots of the quantitative results obtained on RGB images.

**Table 1 entropy-25-01467-t001:** Comparative visual results to GS image.

Original GS Image 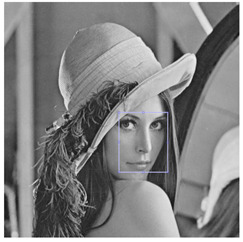						
**Noisy Images**						
σ=0	σ=0.10	σ=0.15	σ=0.20	σ=0.30	σ=0.40	σ=0.50
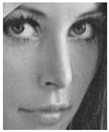	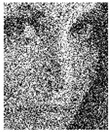	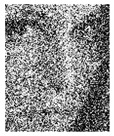	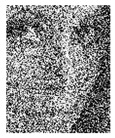	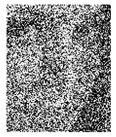	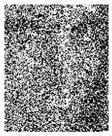	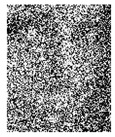
DVA results						
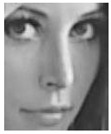	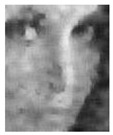	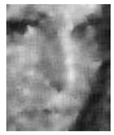	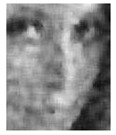	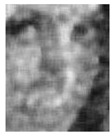	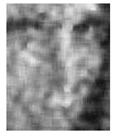	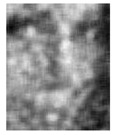
DnCNN results						
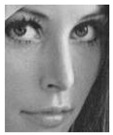	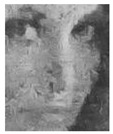	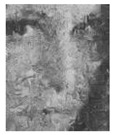	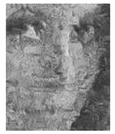	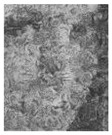	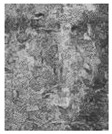	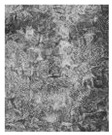
Restormer results						
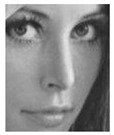	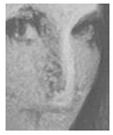	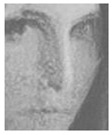	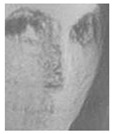	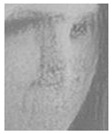	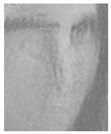	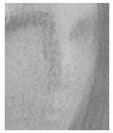
Nafnet results						
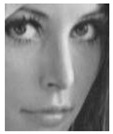	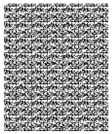	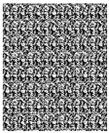	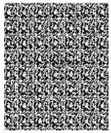	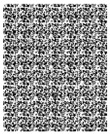	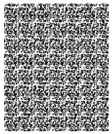	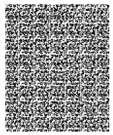

**Table 2 entropy-25-01467-t002:** Comparative visual results to RGB image.

Original RGB Image 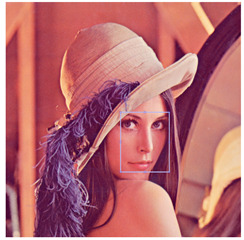						
**Noisy Images**						
σ=0	σ=0.10	σ=0.15	σ=0.20	σ=0.30	σ=0.40	σ=0.50
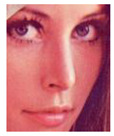	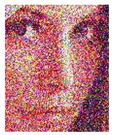	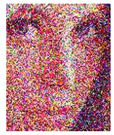	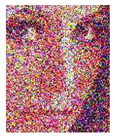	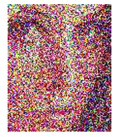	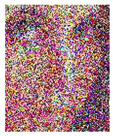	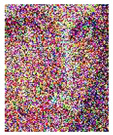
DVA results						
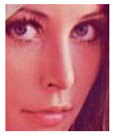	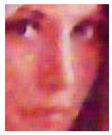	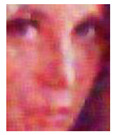	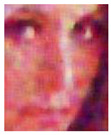	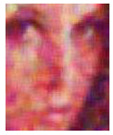	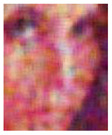	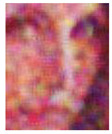
DnCNN results						
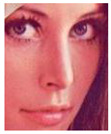	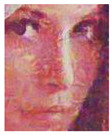	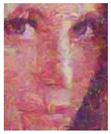	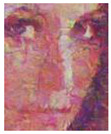	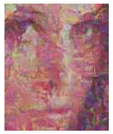	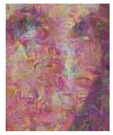	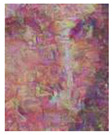
Restormer results						
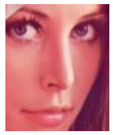	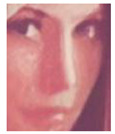	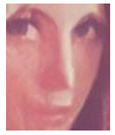	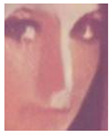	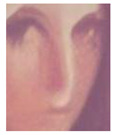	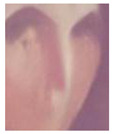	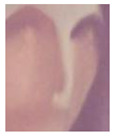
Nafnet results						
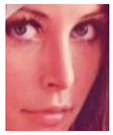	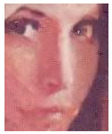	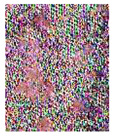	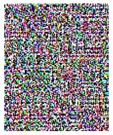	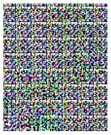	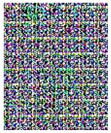	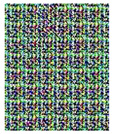

**Table 3 entropy-25-01467-t003:** Comparative results of PSNR in GS images.

GS Image	Density	Noisy Image	DVA	DnCNN	Restormer	Nafnet
Airplane GS	0	inf	26.545	**71.197**	36.987	32.961
	0.10	11.859	**23.729**	22.305	22.137	10.312
	0.15	10.610	**23.014**	20.097	20.818	7.995
	0.20	9.841	**22.378**	18.705	20.128	8.717
	0.30	8.896	**20.938**	16.859	19.132	9.407
	0.40	8.338	**20.474**	15.833	18.476	8.043
	0.50	7.959	**19.312**	15.109	17.937	7.823
Baboon GS	0	inf	17.478	**33.966**	26.414	10.021
	0.10	11.298	19.010	**20.203**	17.560	8.926
	0.15	10.221	18.103	**19.277**	16.634	9.159
	0.20	9.592	18.596	**18.676**	16.003	8.892
	0.30	8.824	**18.222**	17.654	15.294	8.761
	0.40	8.377	**17.913**	16.975	14.827	8.840
	0.50	8.066	**17.702**	16.480	14.476	8.861
Barbara GS	0	inf	23.640	**39.198**	32.285	8.417
	0.10	11.469	**21.795**	21.669	17.472	8.846
	0.15	10.336	**21.309**	20.191	16.120	9.119
	0.20	9.673	**20.119**	19.171	15.245	8.514
	0.30	8.837	**20.160**	17.726	14.150	8.054
	0.40	8.330	**19.672**	16.762	13.450	8.051
	0.50	8.029	**18.975**	16.241	13.083	8.124
Cablecar GS	0	inf	25.853	**67.160**	36.974	31.302
	0.10	12.069	**22.686**	20.751	17.113	7.295
	0.15	10.800	**22.084**	19.047	15.690	6.993
	0.20	9.951	**21.032**	17.636	14.640	7.270
	0.30	8.910	**20.558**	15.981	13.406	7.123
	0.40	8.290	**19.643**	14.945	12.686	6.887
	0.50	7.872	**18.765**	14.216	12.198	6.826
Goldhill GS	0	inf	27.997	**52.056**	39.720	33.700
	0.10	11.595	**24.867**	22.684	17.541	7.818
	0.15	10.450	**23.896**	20.744	15.958	8.031
	0.20	9.722	**23.346**	19.390	14.898	7.954
	0.30	8.857	**22.313**	17.686	13.676	7.718
	0.40	8.335	**21.505**	16.637	12.948	7.716
	0.50	7.971	**20.774**	15.874	12.460	7.640
Lenna GS	0	inf	30.196	**72.566**	38.527	35.414
	0.10	11.383	**24.344**	23.652	18.997	8.645
	0.15	10.284	**23.743**	21.720	17.578	9.051
	0.20	9.619	**22.941**	20.332	16.749	8.815
	0.30	8.825	**21.901**	18.565	15.609	8.394
	0.40	8.350	**21.074**	17.501	14.968	8.531
	0.50	8.049	**20.650**	16.899	14.571	8.566
Mondrian GS	0	inf	20.117	**59.524**	31.921	30.121
	0.10	12.534	**19.672**	18.876	16.526	5.621
	0.15	11.070	**20.003**	17.094	14.994	5.678
	0.20	10.075	**19.170**	15.790	13.970	5.581
	0.30	8.842	**18.086**	14.121	12.713	5.426
	0.40	8.094	**16.578**	13.068	11.969	5.475
	0.50	7.581	**16.204**	12.323	11.446	5.447
Peppers GS	0	inf	25.598	**62.046**	38.161	34.348
	0.10	11.479	**24.303**	23.371	18.504	8.340
	0.15	10.353	**23.010**	21.187	16.975	8.754
	0.20	9.667	**22.402**	19.909	16.064	8.560
	0.30	8.829	**21.752**	18.033	14.940	8.160
	0.40	8.363	**21.193**	17.149	14.347	8.159
	0.50	8.023	**20.383**	16.363	13.838	8.258

**Table 4 entropy-25-01467-t004:** Comparative results of PSNR in RGB images.

RGB Image	Density	Noisy Image	DVA	DnCNN	Restormer	Nafnet
Airplane RGB	0	inf	26.215	**55.638**	36.502	32.961
	0.10	14.576	**24.082**	22.852	23.812	10.312
	0.15	13.342	**23.365**	20.843	22.569	7.995
	0.20	12.526	**22.461**	19.449	21.548	8.717
	0.30	11.525	**21.899**	17.694	20.237	9.407
	0.40	10.922	**21.228**	16.665	19.421	8.043
	0.50	10.503	**19.762**	15.926	18.798	7.823
Baboon RGB	0	inf	21.614	**25.291**	23.442	10.021
	0.10	14.043	19.171	**19.781**	17.699	8.926
	0.15	12.981	18.895	**18.917**	16.758	9.159
	0.20	12.314	**18.704**	18.245	16.122	8.892
	0.30	11.488	**18.475**	17.324	15.377	8.761
	0.40	10.961	**18.144**	16.665	14.828	8.840
	0.50	10.653	**17.850**	16.297	14.521	8.861
Barbara RGB	0	inf	27.412	**39.115**	31.285	29.037
	0.10	14.269	21.742	**21.857**	18.259	16.990
	0.15	13.134	**21.271**	20.416	17.002	8.152
	0.20	12.425	**21.059**	19.426	16.145	8.285
	0.30	11.553	**20.518**	18.128	15.050	7.846
	0.40	11.033	**20.157**	17.285	14.348	7.867
	0.50	10.663	**19.707**	16.726	13.854	8.348
Cablecar RGB	0	inf	22.794	**52.131**	34.426	30.961
	0.10	14.652	**21.977**	20.843	18.035	10.152
	0.15	13.293	**21.563**	18.983	16.419	7.520
	0.20	12.411	**20.120**	17.758	15.419	7.403
	0.30	11.284	**20.164**	16.115	14.106	6.997
	0.40	10.612	**19.757**	15.146	13.304	6.878
	0.50	10.143	**19.036**	14.452	12.725	6.985
Goldhill RGB	0	inf	32.649	**51.974**	36.456	32.535
	0.10	14.323	**23.988**	22.748	19.003	8.023
	0.15	13.149	**23.362**	20.968	17.287	8.134
	0.20	12.392	**23.037**	19.680	16.187	7.666
	0.30	11.501	**22.456**	18.193	14.890	7.438
	0.40	10.927	**21.856**	17.201	14.020	7.585
	0.50	10.558	**21.181**	16.556	13.482	7.853
Lenna RGB	0	inf	28.446	**33.758**	32.538	31.828
	0.10	14.368	**23.799**	23.141	21.068	21.847
	0.15	13.249	**23.332**	21.434	19.475	10.198
	0.20	12.496	**22.966**	20.143	18.344	8.230
	0.30	11.611	**22.467**	18.691	17.022	8.185
	0.40	11.084	**21.703**	17.758	16.191	8.164
	0.50	10.707	**21.152**	17.063	15.629	8.189
Mondrian RGB	0	inf	17.688	**36.324**	29.113	28.609
	0.10	14.728	16.729	**17.404**	16.621	15.873
	0.15	13.072	**16.465**	15.700	14.978	14.440
	0.20	11.976	**15.927**	14.560	13.850	13.526
	0.30	10.568	**15.098**	13.054	12.432	12.291
	0.40	9.690	**14.841**	12.086	11.545	11.420
	0.50	9.070	**15.039**	11.391	10.917	10.330
Peppers RGB	0	inf	33.057	**48.801**	34.615	32.112
	0.10	14.519	**24.496**	22.653	19.361	19.103
	0.15	13.324	**23.756**	20.752	17.669	17.418
	0.20	12.540	**23.349**	19.468	16.594	16.102
	0.30	11.565	**22.606**	17.837	15.310	7.490
	0.40	10.974	**21.553**	16.868	14.491	7.657
	0.50	10.569	**20.784**	16.179	13.942	7.667

**Table 5 entropy-25-01467-t005:** Visual and quantitative results obtained by DVA in HD images.

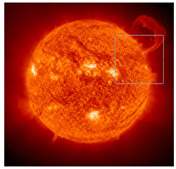			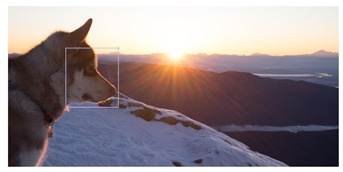		
Sun 2100 × 2034					
σ=0	σ=0.10	σ=0.20	σ=0.30	σ=0.40	σ=0.50
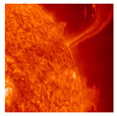	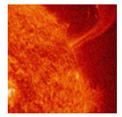	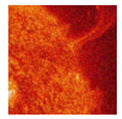	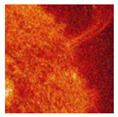	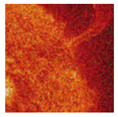	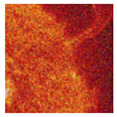
	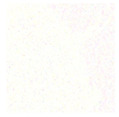	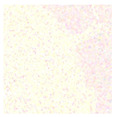	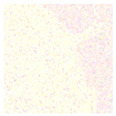	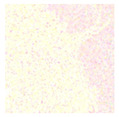	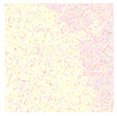
ERGAS = 5169.806	ERGAS = 10,965.422	ERGAS = 13,395.159	ERGAS = 15,276.500	ERGAS = 17,736.873	ERGAS = 18,296.674
MSE = 21.131	MSE = 124.536	MSE = 249.594	MSE = 380.183	MSE = 567.533	MSE = 699.633
PSNR = 34.882	PSNR = 27.178	PSNR = 24.158	PSNR = 22.331	PSNR = 20.591	PSNR = 19.682
RASE = 0	RASE = 1498.244	RASE = 1902.722	RASE = 2190.058	RASE = 2530.487	RASE = 2639.515
RMSE = 4.597	RMSE = 11.160	RMSE = 15.799	RMSE = 19.498	RMSE = 23.823	RMSE = 26.451
SAM = 0.072	SAM = 0.273	SAM = 0.390	SAM = 0.448	SAM = 0.489	SAM = 0.523
SSIM = 0.994	SSIM = 0.964	SSIM = 0.926	SSIM = 0.896	SSIM = 0.867	SSIM = 0.842
UQI = 0.782	UQI = 0.558	UQI = 0.512	UQI = 0.499	UQI = 0.490	UQI = 0.484
Dog 6000 × 2908					
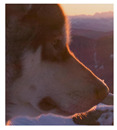	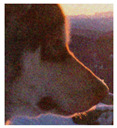	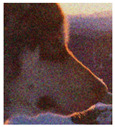	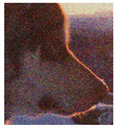	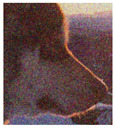	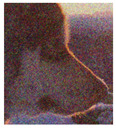
	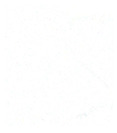	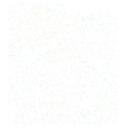	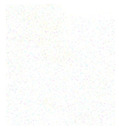	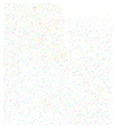	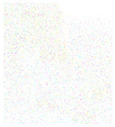
ERGAS = 5624.483	ERGAS = 11,456.096	ERGAS = 10,623.462	ERGAS = 10,393.671	ERGAS = 9919.464	ERGAS = 10,406.266
MSE = 217.856	MSE = 362.834	MSE = 441.465	MSE = 566.388	MSE = 610.187	MSE = 763.037
PSNR = 24.749	PSNR = 22.534	PSNR = 21.682	PSNR = 20.6	PSNR = 20.276	PSNR = 19.305
RASE = 806.958	RASE = 1652.544	RASE = 1530.997	RASE = 1496.294	RASE = 1427.232	RASE = 1496.917
RMSE = 14.76	RMSE = 19.048	RMSE = 21.011	RMSE = 23.799	RMSE = 24.702	RMSE = 27.623
SAM = 0.022	SAM = 0.078	SAM = 0.089	SAM = 0.099	SAM = 0.113	SAM = 0.131
SSIM = 0.936	SSIM = 0.773	SSIM = 0.711	SSIM = 0.665	SSIM = 0.623	SSIM = 0.588
UQI = 0.986	UQI = 0.936	UQI = 0.948	UQI = 0.953	UQI = 0.956	UQI = 0.951

## Data Availability

Not applicable.

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
