# Peer review of "Denoising Vanilla Autoencoder for RGB and GS Images with Gaussian Noise"

_entropy, 2023, doi:10.3390/e25101467_

Round 1
Reviewer 1 Report
The language of the paper should be revised and some statements need rephrasing.
The acronyms in the abstract should be disguised, as in reading thecabstract we need to know in advance what they are. The abstract should include some quantitative results here achieved.
Regarding the denoising problem here analysed, I suggest to mention the fuzzy logic approach.
See (suggested citation):
M. Versaci, F. C. Morabito and G. Angiulli, "Adaptive Image Contrast Enhancement by Computing Distances into a 4-Dimensional Fuzzy Unit Hypercube," in IEEE Access, vol. 5, pp. 26922-26931, 2017, doi: 10.1109/ACCESS.2017.2776349.
Autoencoders are quite relevant methodologies well suited for the specific application; for this reason, you should make an additional effort to well describe the methodologies and compare them.
The captions of all of the figures need to be improved for making easy to understand them and their importance. In particular, Figure 4 can be improved.
Tables 5 and 6 are Figures. They need a suitable caption, that is able to describe what they represent.
The relative merits of the different metrics propose4d should be highlighted.
It can be easily improved.
Reviewer 2 Report
This article presents a convolutional auto-encoder for image denoising. The method is called DVA and is presented as new, however the methodology is lacking novelty. Also, there are several examples of writing in English. It seems that the document was drafted in haste.
Eg. "many Sometimes" in the abstract ?
in line 29, "learning methods deep learning have ... " should be rephrased.
in line 33, "an image that I did not know just by ... " is not clear
in the caption of figure 2, the word "Diference" should be "Figure 2. Difference"
in line 123, sigma = 0.20 however in figure 2, \sigma=20 which is the correct one?
In the architecture of figure 2, the channels are processed in 3 different streams. I am wondering if the outputs of the noisy channels are not correlated ? the partitions with similar colors from noisy images are visually similar?
Finally, the proposed model has been already presented by Keras tutorial, see : https://keras.io/examples/vision/autoencoder/
The proposed model is quite similar to Keras tutorial, see : https://keras.io/examples/vision/autoencoder/
Reviewer 3 Report
This paper presents a new methodology called Denoising Vanilla Autoencoder (DVA) for enhancing RGB and grayscale (GS) images by suppressing Gaussian noise. The proposed methodology utilizes unsupervised neural networks and demonstrates superior performance compared to other existing noise suppression techniques. The authors support their claims with numerical results obtained from a validation set, as well as visual evidence.
Review:
General Comments: The paper addresses an important problem in computer vision, namely, the denoising of images corrupted by Gaussian noise. The proposed DVA methodology shows promising results in terms of noise suppression. However, there are certain aspects that need improvement in order to strengthen the paper's contribution. Additionally, the authors should consider citing relevant works to establish the context and enhance the significance of their research.
-
Introduction and Motivation: The introduction provides a clear overview of the problem statement and its significance in computer vision. The authors adequately highlight the importance of noise suppression for robust image analysis. However, it would be beneficial to include more specific examples or applications where noise suppression is crucial. Furthermore, the authors should consider explaining the motivation behind using unsupervised neural networks for denoising and how it differs from supervised or semi-supervised approaches.
-
Related Work: The paper lacks a comprehensive review of the related literature on image denoising techniques. It is essential to discuss existing methods, such as traditional filters (e.g., median, Wiener), deep learning-based approaches (e.g., denoising autoencoders, generative adversarial networks), and their strengths and limitations. To address this gap, I recommend the authors cite the following papers:
-
"A Comprehensive Survey of Scene Graphs: Generation and Application": This paper provides insights into the broader field of computer vision and scene understanding. Scene graphs are widely used for various tasks, including image denoising. By citing this work, the authors can establish the relevance of their research in the context of scene understanding.
-
"TN-ZSTAD: Transferable Network for Zero-Shot Temporal Activity Detection": Although the cited paper focuses on a different problem, it presents transferable network architectures that can be relevant for image denoising. The authors can draw parallels between transfer learning in activity detection and their proposed DVA methodology, highlighting the potential for knowledge transfer across domains.
-
"Video Pivoting Unsupervised Multi-Modal Machine Translation": This paper explores unsupervised learning methods for video translation. While the task differs from denoising, it provides insights into the unsupervised learning paradigms relevant to the authors' work. By citing this paper, the authors can demonstrate the broader applicability of unsupervised neural networks and establish connections between related fields.
-
"ZeroNAS: Differentiable Generative Adversarial Networks Search for Zero-Shot Learning": This paper introduces a differentiable approach for generating adversarial networks. Although its focus is on zero-shot learning, the proposed methodology can be relevant for the denoising task. Citing this paper would enable the authors to showcase the potential of differentiable networks in noise suppression.
- Methodology and Experiments: The paper provides a brief overview of the proposed DVA methodology, but the technical details are lacking. It is crucial to describe the architecture of the vanilla autoencoder, the loss function used for training, and any specific modifications or enhancements made for denoising. Additionally, the authors should clarify the rationale behind choosing a vanilla autoencoder and discuss its advantages over other architectures.
The experimental section should provide more details on the dataset used for validation, including the size, composition, and sources. Furthermore, the authors should explain the implementation details, such as hyperparameters and training setup, to ensure reproducibility. It would be helpful if the authors report quantitative metrics, such as Peak Signal-to-Noise Ratio (PSNR) or Structural Similarity Index (SSIM), to compare the performance of the proposed DVA methodology with existing methods.
- Conclusion: The conclusion effectively summarizes the key findings and highlights the superiority of the proposed DVA methodology. However, the authors should consider discussing potential limitations and avenues for future research. Additionally, they should emphasize the significance of their work by connecting it to real-world applications or potential downstream tasks in computer vision.
In summary, the paper presents an interesting approach for denoising RGB and GS images using a vanilla autoencoder. However, the authors should address the mentioned concerns and incorporate the suggested citations to improve the overall quality and impact of the paper.
NA
Round 2
Reviewer 1 Report
I am now in favor to publish it.
I am now in favor to publish it.